# Histone Demethylase JMJD2D: A Novel Player in Colorectal and Hepatocellular Cancers

**DOI:** 10.3390/cancers14122841

**Published:** 2022-06-08

**Authors:** Qiang Chen, Kesong Peng, Pingli Mo, Chundong Yu

**Affiliations:** 1State Key Laboratory of Cellular Stress Biology, Innovation Center for Cell Biology, School of Life Sciences, Xiamen University, Xiamen 361102, China; 21620170155389@stu.xmu.edu.cn; 2School of Medicine, Ningbo University, Ningbo 315021, China; 3Department of Cellular and Genetic Medicine, School of Basic Medical Sciences, Fudan University, Shanghai 200032, China; kesongpeng@fudan.edu.cn

**Keywords:** JMJD2D, KDM4D, H3K9me3, epigenetic, colorectal cancer, hepatocellular cancer

## Abstract

**Simple Summary:**

Histone demethylase JMJD2D is a multifunctional epigenetic factor coordinating androgen receptor activation, DNA damage repair, DNA replication, cell cycle regulation, and inflammation modulation. JMJD2D is also a well-established epigenetic facilitator in the progression of multiple malignant tumors, especially in colorectal cancer (CRC) and hepatocellular cancer (HCC). This review aims to summarize the mechanisms of JMJD2D in promoting CRC and HCC progression, which provides novel ideas for targeting JMJD2D in oncotherapy. JMJD2D promotes gene transcription by reducing H3K9 methylation and serves as a coactivator to enhance the activities of multiple carcinogenic pathways, including Wnt/β-catenin, Hedgehog, HIF1, JAK-STAT3, and Notch signaling; or acts as an antagonist of the tumor suppressor p53.

**Abstract:**

Posttranslational modifications (PTMs) of histones are well-established contributors in a variety of biological functions, especially tumorigenesis. Histone demethylase JMJD2D (also known as KDM4D), a member of the JMJD2 subfamily, promotes gene transcription by antagonizing H3K9 methylation. JMJD2D is an epigenetic factor coordinating androgen receptor activation, DNA damage repair, DNA replication, and cell cycle regulation. Recently, the oncogenic role of JMJD2D in colorectal cancer (CRC) and hepatocellular cancer (HCC) has been recognized. JMJD2D serves as a coactivator of β-catenin, Gli1/2, HIF1α, STAT3, IRF1, TCF4, and NICD or an antagonist of p53 to promote the progression of CRC and HCC. In this review, we summarize the molecular mechanisms of JMJD2D in promoting the progression of CRC and HCC as well as the constructive role of its targeting inhibitors in suppressing tumorigenesis and synergistically enhancing the efficacy of anti-PD-1/PD-L1 immunotherapy.

## 1. Introduction

Genetic materials are tightly condensed in a core of positively charged histones that surround negatively charged DNA wraps. The theory of epigenetics was first proposed by Conrad h. Waddington in 1942, which has been recognized as epigenetic modifications mediating cellular phenotype or gene expression through DNA or histone covalent modifications, chromatin remodeling, non-coding RNA, etc., without altering the sequence of DNA [1,2]. The core histones constituting nucleosomes contain a Lys- and Arg-rich tail or side chain and are extensively modulated by posttranslational modifications (PTMs) [3]. The acetylation, methylation, and phosphorylation modifications of histone have been widely reported, while it can also be modified by other means such as O-acetyl glycosylation, formylation, and ADP-ribosylation [4,5,6,7,8].

Cancer progression is closely related to the changes in histone PTMs, and the abnormal control of PTMs-related enzymes, including histone methylase, demethylase, acetylase, and acetyltransferase, is an important predisposing factor for tumors [9]. These epigenetic changes may silence multiple tumor suppressor genes or activate oncogenes, leading to the reprogramming of oncogenes in the genome. Allfrey et al. first reported that histone methylation is related to gene transcriptional regulation [10], and then the modifications on Lys, Arg, Ser, Thr, Tyr, and His residues have been reported [11]. The most widely studied modified site is Lys, which can be monomethyl, dimethyl, or trimethyl (i.e., me1, me2, or me3). There is a relatively large lysine methyltransferase family [12]; however, the same type of methylation modification cannot be catalyzed by all enzymes, which have substrate sequence preference. For example, EZH2 can mediate the monomethylation, dimethylation, and trimethylation of H3K27, while G9a (EHMT2) and MMSET (NSD2) mediate the monomethylation and dimethylation of H3K9 and H3K36, respectively, but not trimethylation [13].

Histone methylation modification was once considered to be an irreversible process, which is overturned by the discovery of LSD1 and KDM demethylase family with Jumonji C (JmjC) domain [14,15,16]. Unlike LSD1, KDMs are oxygenases that target multiple sites including H3K9, H3K27, and H3K36, and utilize 2-ketoglutarate (2-OG) and Fe^2+^ as cofactors to realize the demethylation of substrates through the mechanism of generating formaldehyde-free radicals [17]. The KDM demethylase JMJD2D (also known as KDM4D) belongs to JMJD2 subfamily that contains five members (JMJD2A-E) and can recognize the dimethyl and trimethylation of H3K9 and H3K36, as well as the trimethylation of H1.4K26 [9,18]. H3K9 and H1.4K26 trimethylations are related to transcriptional inhibition or heterochromatin formation, while H3K36 methylation is related to the activation of genes; and methylation modification at these sites is not only involved in gene transcription control but also related to DNA replication and repair [19,20]. Therefore, the dynamic transition of histone methylation regulated by the JMJD2 family may have a far-reaching impact on various cellular physiological activities, especially on tumorigenesis. The research on the function of JMJD2A, JMJD2B, and JMJD2C is relatively comprehensive, while that of JMJD2D is ongoing, but JMJD2E is unformed. Here, we focus on JMJD2D, whose oncogenic role has recently been recognized for its diverse expression patterns in several tissues and pathological states.

## 2. JMJD2D Promotes Gene Transcription by Antagonizing H3K9 Methylation

KDM4/JMJD2 histone demethylase family members are very similar in overall protein structure, which contains the featured JmjN and JmjC domain. JMJD2A, JMJD2B, and JMJD2C contain two plant homeodomains (PHD) and two tudor domains, while JMJD2D and JMJD2E are smaller and lack of the C-terminal PHD and tudor domains [9,18,21] (Figure 1A). Intriguingly, Shim et al. demonstrated that the truncation mutants of JMJD2A and JMJD2C that lack the tudor or both the tudor and PHD domains can still demethylate H3K9 and H3K36 [22]. Methylation of some lysine residues in H3 histones can activate or inactivate the transcriptional activity of genes, including H3K4, H3K9, H3K17, H3K27, H3K36, and H3K79 [23], while the dimethylation or trimethylation of H3K9 (H3K9me2/3) and H3K27 (H3K27me2/3) primarily associate with heterochromatin and gene repression [24,25,26,27,28,29]. The full-length JMJD2D, which localizes in the human chromosome 11q21 [18,30], demethylates the histone residues H3K9me2/3 and H1.4K26me3 to the monomethyl state, although H3K9me2/3 is the preferred substrate [9,31]. The JmjC domain of JMJD2D is the catalytic core, which facilitates a dioxygenase reaction requiring Fe^2+^, O_2_, and 2-oxoglutarate (2-OG) to demethylate histones. The oxygen atom is incorporated into the methyl group during oxidative decarboxylation; subsequently, an unstable intermediate imine product is formed and then converted to formaldehyde and demethylated product (Figure 1B) [32]. As expected, the JmjC-domain-lacking JMJD2 protein is deficient in demethylation activity [22]. The histidine 192 on the JmjC domain of JMJD2D is essential for its histone demethylase function and mutation of this residue severely compromises the catalytic activity of JMJD2D, leading to the loss of its ability to stimulate the mouse mammary tumor virus (MMTV) promoter [33]. The serine 200 on the JmjC domain is also essential for the histone demethylase function of JMJD2D and mutating this residue to methionine (S200M) generates a demethylase-dead mutant [34,35]. The JmjN domain of JMJD2D may be responsible for the integrity of structure and serves as a dimerization interface [9,18,21,36]. Deletion of the JmjN domain also abrogates the histone demethylase activity in the cell, due to the nuclear exclusion of JMJD2D [22].

## 3. JMJD2D Has Multiple Biological Functions

As mentioned earlier, JMJD2D promotes gene transcription by antagonizing H3K9 methylation, while it can serve as a coactivator to enhance the transcriptional activities of multiple transcription factors (TFs). JMJD2D is a well-established epigenetic regulator in a variety of biological functions, including androgen receptor (AR) activation, DNA damage repair, DNA replication, cell cycle regulation, inflammation modulation, and tumorigenesis promotion (Figure 2).

### 3.1. JMJD2D Facilitates the Functions of Androgen Receptor

JMJD2D first attracted attention as an AR activator [33], and the study of its biological function has been ongoing since it was identified. AR is a transcription factor that plays a pivotal role in the development of prostate cancer. Shin and colleagues first reported that JMJD2D is a novel AR coactivator, which can form a complex with ligand-bound AR via its C-terminus to promote the progression of prostate cancer [33]. Androgen is also an important regulator for trophoblast differentiation and placental development, JMJD2D facilitates this process via forming the AR-KDMs complex and promoting the placental androgen signaling to regulate placental VEGFA expression [37]. JMJD2D, which is highly expressed in testis and strongly associated with testis morphology traits [38], demethylates H3K9me3 during spermatogenesis in mice; however, the fertility of JMJD2D-deficient mice is not undulated although H3K9me3 accumulates significantly in round spermatids [39].

### 3.2. JMJD2D Plays a Key Role in DNA Damage Repair, DNA Replication, and Cell Cycle Regulation

DNA damage is one of the hallmarks of cancer that leads to genomic instability, in which the double-strand breaks (DSB) are the most deleterious damage forms and H3K9me3 is one of the epigenetic barriers to DSB repair [40,41]. Previously, JMJD2B demethylase was identified as a DNA damage response protein, which can be recruited to DNA damage tracks induced by laser micro-irradiation in a demethylase-dependent manner [40]. Khoury-Haddad et al. also reported that JMJD2D demethylase, which can be rapidly recruited to DNA damage regions, is required for DSB repair, while the demethylase activity of JMJD2D is dispensable for the accumulation at DNA damage regions, but its C-terminal region is essential [34]. Intriguingly, Khoury-Haddad et al. demonstrated in another study that JMJD2D–RNA interaction is required for the recruitment of JMJD2D to DNA damage sites, and the efficient demethylation of H3K9me3 by JMJD2D is essential [42].

H3K9me3 is associated with transcriptional silencing in heterochromatin, while it can also inactivate the enhancers of the cell-type-specific gene to participate in the precise regulation of gene expression, and the demethylation of H3K9me3 by JMJD2D demethylase is essential in this process [43,44]. Furthermore, sufficient evidence showed that JMJD2D is involved in DNA replication and cell cycle regulation [45,46,47]. H3K9me3 is associated with the silencing of cell cycle genes that are essential for cardiac myocyte (CM) cell cycle exit; while overexpression of JMJD2D specifically attenuates H3K9me3 levels and increases the expression of cell-cycle-associated proteins in CM, resulting in CM hyperplasia [47]. Again, Wu et al. reported that JMJD2D combines with DNA replication-related proteins via its JmjC domain in a demethylation activity independent manner during G1 and S phases of the cell cycle, which promotes the initiation and elongation of DNA replication [45]. Although the binding of JMJD2D does not depend on its demethylase activity, reducing the level of H3K9me3 can rescue the DNA replication defect in JMJD2D-deficient cells, indicating that the demethylase activity of JMJD2D may be essential [45]. 

### 3.3. JMJD2D Modulates the Inflammatory Response

In addition to the above-mentioned JMJD2D-modulated biological activities, other studies have shown that JMJD2D may be involved in immune and inflammatory responses, and it also plays a critical role in the neurogenesis of dynamic hippocampal dentate gyrus [48,49]. Jin and colleagues reported that JMJD2D is involved in the regulation of proinflammatory cytokines interleukin 12 and 23 (IL-12 and IL-23), which are regulators of immune response and inflammation [48]. TRAF-binding protein domain (Trabid, also known as Zranb1), a deubiquitinase that preferentially hydrolyzes lysine 29 (K29)- and K33-linked ubiquitin chain [50,51], is a crucial regulator of TLR-stimulated IL-12 and IL-23 expression [48]. LPS stimulation upregulates the expression of JMJD2D and promotes its recruitment to the IL12a, IL12b, and IL23a promoters [48,52]. Trabid effectively cooperates with JMJD2D to promote the expression of these inflammatory factors by preferentially reducing the K29 ubiquitination of JMJD2D and stabilizing JMJD2D [48]. The autophagic degradation of JMJD2D in modulating inflammatory response has also been reported. Tripartite motif 14 (TRIM14), which belongs to one of the most dominant E3 ligase families, is a critical regulator of type I IFN signaling [53] and an important oncogene [54,55]. TRIM14 decreases the H3K9me2/3 level to facilitate the transcription of IL-12 and IL-23 by reducing the K63-linked ubiquitination of JMJD2D and subsequently prevents its selective autophagic degradation [56].

## 4. JMJD2D Is a Pivotal Activator of Colorectal and Hepatocellular Cancers

Epigenetic dysregulation is a predisposing factor for tumorigenesis, in which the deregulation of H3K9me3 is associated with oncogene activation or tumor suppressor silencing. Many studies have demonstrated that abnormal expression of JMJD2 demethylases including JMJD2A, JMJD2B, and JMJD2C, promotes the progression of lymphatic, prostate, lung, esophageal, and breast cancers [57,58,59,60,61,62]. Recently, the oncogenic role of JMJD2D has been recognized in colorectal cancer (CRC) and hepatocellular cancer (HCC).

### 4.1. JMJD2D Promotes the Progression of Colorectal Cancer

The occurrence of CRC involves the abnormal activation of multiple cellular signaling, including Wnt/β-catenin, NF-κB, Hedgehog, PI3K/Akt, JAK/STAT, Notch, and MAPK signal pathways [63,64]. JMJD2D has been reported to promote the progression of CRC by participating in one or more of these signaling modulations [35,65,66,67]. A previous study has reported the potential carcinogenesis of JMJD2D in HCT116 cells, but its special mechanism and in vivo role are not clear [68]. The exact oncogenic role of JMJD2D in CRC is recognized by Peng and colleagues [35]. JMJD2D is highly expressed in tumor tissues of CRC patients; knockdown or knockout of JMJD2D in CRC cells can suppress their proliferation in vitro and inhibit the growth of xenograft tumors and the metastasis to lung or liver in vivo; similarly, the formation of AOM/DSS-induced CRC is also attenuated in JMJD2D^−/−^ mice [35]. Abnormal Wnt/β-catenin signaling is a famous driver of carcinogenesis in multiple tumors, especially in CRC. β-catenin is mutated in about 90% of CRC patients and plays a transcriptional role in the expression of various oncogenes, including Myc, Cyclin D1, and matrix metalloproteinase (MMPs) [69,70]. JMJD2D is a facilitator for β-catenin expression: JMJD2D is recruited to the promoter of β-catenin to reduce its surrounding H3K9me3, which facilitates the transcription of β-catenin. On the other hand, JMJD2D serves as a co-activator for the transcriptional activity of β-catenin: JMJD2D combines with β-catenin via its JmjC domain, subsequently co-recruiting to the promoters of β-catenin target genes, such as Myc, Cyclin D1, MMP2, and MMP9, to release the H3K9me3 blocking on these promoters [35]. Intriguingly, the recruitment of β-catenin to the promoters of its target genes may also be modulated by JMJD2D, as demonstrated by attenuated β-catenin recruitment on its target gene promoters in JMJD2D-deficient CRC cells [35].

Since JMJD2D is frequently upregulated in CRC [35], it is important to reveal its underlying mechanisms. A novel transcriptional regulatory mechanism of JMJD2D has been elucidated [66]. Two NF-κB response elements are identified on the JMJD2D promoter [66]. Activation of NF-κB signaling by TNFα induces JMJD2D expression, whereas inhibition of NF-κB signaling by aspirin reduces JMJD2D expression [66], suggesting that JMJD2D expression is regulated by NF-κB signaling. TNFα released by immune cells (such as macrophages) in DSS-induced colitis also induces the expression of JMJD2D by activating NF-κB signaling [66,67]. Upregulated JMJD2D promotes the renewal of colonic epithelial cells by activating Hedgehog (Hh) signaling, but it also facilitates the malignant proliferation of cancerous cells [71]. Gli2, which is transcriptionally modulated by activated Hh signaling, serves as a main transcriptional activator of multiple Hh-target oncogenes, such as Bcl-2, Cyclin D1, Slug, and β-catenin. The expression of Gli2 and these Hh-target genes is suppressed in JMJD2D-deficient cells or tissues [66,72], suggesting that JMJD2D can enhance Hh signaling. Mechanistically, JMJD2D serves as a coactivator of Gli2 and is co-recruited with Gli2 to the promoter of Hh-target genes by directly binding to Gli2, facilitating the transcription of these genes by demethylating H3K9me3 [66].

Hypermetabolism provides essential assistance in facilitating cancerous cell malignant proliferation and metastasis, in which the glycolysis meets the urgent nutrition supply [73]. The tumor microenvironment is hypoxic, and cancerous cells undergo various biological activities in response to hypoxia, of which is that JMJD2D co-activates multiple cellular signals to promote glycolysis in tumor cells [65]. Hypoxia-inducible factor 1 (HIF1) is a heterodimer, which contains a HIF1α regulatory subunit and a HIF1β constitutive subunit, that plays a critical role in facilitating the transcription of a panel of genes related to tumor proliferation, metastasis, angiogenesis, and glycolysis, such as lactate dehydrogenase A (LDHA), phosphoglycerate kinase 1 (PGK1), and monocarboxylate transporter 4 (MCT4) [74,75,76]. The glycolysis and the expression of its related genes (e.g., HIF1α, HIF1β, PGK1, LDHA, and MCT4) are attenuated in JMJD2D-deficient CRC cells, suggesting that JMJD2D can regulate glycolysis via the HIF1 signaling pathway [65]. Sufficient evidence shows that the protein synthesis of HIF1α is modulated by activated mTOR and MAPK signaling [77,78]. The expression of mTOR can be facilitated by the transcriptional complex formed by JMJD2D and SOX9, thereby increasing HIF1α translation. JMJD2D also promotes the expression of HIF1β by cooperating with transcription factor c-Fos, and the demethylation of H3K9me3 by JMJD2D is essential [65]. Furthermore, JMJD2D can serve as a coactivator of HIF1α to promote the transcription of glycolysis-related genes such as PGK1, LDHA, and MCT4 via demethylating H3K9me3 on the promoters of these genes [65]. Therefore, JMJD2D can promote CRC glycolysis and progression by activating HIF1 signaling via multiple mechanisms [65].

The inefficient immune surveillance is also a key factor in tumorigenesis, in which the inactivity or exhaustion of T cells plays a critical role. Programmed death receptor-1 (PD-1) is one of the important immune checkpoints and serves as an eliminator for effector T cell activation [79]. Its ligand PD-L1 is highly expressed in most tumor cells and antagonizes effector T cells by triggering PD-1-modulated inhibitory signaling [80]. The role of JMJD2D in promoting PD-L1 expression and CRC immune escape was identified recently [67]. Previous studies have reported that JMJD2D is highly expressed in tumor tissues of CRC patients [35,67]. PD-L1 expression is also upregulated in human CRC specimens and positively correlated with JMJD2D expression [67]. Consistently, downregulation of JMJD2D reduces PD-L1 expression in CRC cells [35,67], indicating that JMJD2D can regulate PD-L1 expression. Knockout of JMJD2D in mouse CRC cells inhibits CRC growth in both normal immune C57BL/6 mice and immunocompromised nude mice [67]; however, the proportion of tumor reduction in normal immune C57BL/6 mice is higher than that in immunocompromised nude mice, most likely due to the increased contribution of tumor-infiltrating CD8^+^ T cells [67]. High expression of JMJD2D in CRC could indirectly inhibit the activation and effector function of tumor-infiltrating CD8^+^ T cells by upregulating PD-L1 expression [67]. IFNγ is an inflammatory factor released by immune cells; however, it also serves as a trigger of PD-L1 upregulation by activating JAK-STAT signaling in tumor cells, resulting in the antagonism of CD8^+^T cell surveillance [81]. IFNγ receptor (IFNGR) is a dimeric complex composed of IFNGR1 and IFNGR2, which recruits and activates JAKs kinases to transmit the activation signals [82,83]. JMJD2D can facilitate and maintain the IFNγ-induced activation signals by cooperating with SP-1 transcription factor to promote IFNGR1 transcription [67]. Sufficient evidence indicates that activated JAK-STAT signaling modulates the expression of PD-L1, in which STAT3 serves as a critical accelerator in facilitating PD-L1 transcription; and IRF1, which is regulated by STAT3, also plays an important role in the transcriptional regulation of PD-L1 [82,83]. Of note, JMJD2D can cooperate with the STAT3-IRF1 axis to further enhance the signal of PD-L1 transcription triggered by IFNγ, which depends on the binding of JMJD2D to transcription factors and the demethylase activity of JMJD2D as mentioned earlier [35,65,66,67].

An overview of the mechanisms by which JMJD2D promotes CRC progression is proposed (Figure 3). Briefly, intestinal inflammation can trigger immune cells to release TNFα to upregulate JMJD2D in colorectal cells by activating NF-κB signaling, and high levels of JMJD2D can cooperate with Gli2 and β-catenin to promote the expression of various oncogenes, including Gli1/2, Bcl-2, Slug, β-catenin, c-Myc, Cyclin D1, and MMP2/9. JMJD2D can also cooperate with SOX9 and c-Fos to induce HIF1α/β expression and serves as a coactivator for HIF1α to promote the transcription of tumor glycolysis-related genes. Importantly, JMJD2D can facilitate IFNγ-triggered PD-L1 upregulation, in which JMJD2D cooperates with STAT3 and IRF1 to promote PD-L1 transcription, and subsequently suppresses the activation and effector function of CD8^+^ T cells by activating PD-1/PD-L1 inhibitory signals.

### 4.2. The Pivotal Role of JMJD2D in the Pathogenesis of Hepatocellular Cancer

Hepatic fibrosis is not only the median stage of hepatic cirrhosis but also the pathogenesis of primary HCC [84], in which the activation of the epigenetic mechanism in hepatic stellate cells (HSCs) is an important inducer [85]. JMJD2D is remarkably upregulated in activated HSC and fibrotic liver tissues; knockdown of JMJD2D can inhibit and reverse the progression of hepatic fibrosis in the CCl_4_-induced mouse fibrosis model [85]. Mechanistically, JMJD2D contributes to HSC activation and hepatic fibrosis progression by facilitating the transcription of toll-like receptor 4 (TLR4) through H3K9me2/3 demethylation, subsequently activating TLR4/NF-κB signaling in HSC [85]. Hepatic steatosis is also one of the mechanisms of pathogenesis of primary HCC, in which the abnormal or excessive differentiation of mesenchymal stem cells (MSC) to adipocytes may be a potential risk factor [86,87,88]. Intriguingly, JMJD2D has been reported as a critical regulator for the adipogenic differentiation of C3H10T1/2 MSCs [89]. JMJD2D interacts with NFIB and MLL1 to form a regulatory complex to participate in activating the transcription of key adipogenic regulators (e.g., Pparg and Cebpa), and the demethylation of H3K9me3 at the Cebpa and Pparg promoters by JMJD2D is essential for promoting adipogenic differentiation [89].

The oncogenic role of JMJD2D in HCC was recently identified [90,91]. JMJD2D is highly expressed in tumor tissues of HCC patients, which promotes the initiation and progression of HCC by inhibiting p53 signaling [90]. P53, a critical tumor suppressor that is silenced or mutated in the most common cancers, plays fundamental and multifaceted roles in tumor suppression [92,93,94,95]; p53 induces the cell cycle arrest at the G1/S boundary and triggers cell apoptosis by initiating the transcription of p21 and PUMA [96]. JMJD2D can inhibit the recruitment of p53 to the promoters of p21 and PUMA by directly binding to the DNA binding domain of p53, resulting in the silencing of p21 and PUMA and oncogenesis (Figure 4) [90]. In general, most biological functions of JMJD2D depend on its demethylase activity, and H3K9me2/3 is the preferred substrate [9,31]; however, it seems that JMJD2D downregulates the expression of p21 and PUMA by inhibiting p53 recruitment to the promoters, independent of its demethylase activity [90].

It is well-known that the existence of cancer stem-like cells (CSCs) is a crucial inducer for tumorigenesis, which promotes tumor heterogeneity, metastasis, and recurrence [97,98]. Liver cancer stem-like cells (LCSC) play an essential role in maintaining the progression of HCC, in which EpCAM and SOX9 are critical [99,100,101]. The role of JMJD2D in maintaining LCSCs has been revealed [91]. The self-renewal of JMJD2D-deficient LCSCs is attenuated in vitro, and LCSC-derived tumor initiation, progression, and lung metastasis in vivo are also suppressed when JMJD2D is downregulated [91]. JMJD2D promotes the self-renewal of LCSCs by enhancing the expression of EpCAM and SOX9. Sufficient evidence reveals that the self-renewal of CSCs is modulated by a variety of cellular signals, such as Wnt/β-catenin, TGF-β, Hedgehog, and Notch signaling [102,103,104]. JMJD2D interacts with β-catenin/TCF4 and NICD to facilitate their recruitment to the promoters of EpCAM and SOX9 [91]; JMJD2D promotes the transcription of EpCAM and SOX9 by demethylating H3K9me3 in a demethylase-dependent manner (Figure 4) [91].

### 4.3. JMJD2D and Other Tumors

The mechanisms by which JMJD2D promotes the pathogenesis of CRC and HCC have been preliminarily clarified, while the facilitator role of JMJD2D in other malignancies has also been reported, including gastrointestinal stromal tumor (GIST), acute myeloid leukemia (AML), renal cell carcinoma (RCC), and esophageal squamous cell carcinoma (ESCC) [105,106,107,108]. JMJD2D is found to be highly expressed in GIST tumor tissues, and overexpression of JMJD2D can promote the proliferation and angiogenesis of GIST cells both in vitro and in vivo, which is reversed in JMJD2D-deficient GIST cells [105]. Abnormal HIF/VEGFA signaling is a key facilitator of tumor angiogenesis, in which HIF1β plays a critical role in a HIF1α-independent manner [109,110]. Hu and colleagues reported that JMJD2D promotes the progression of GIST by activating HIF1β/VEGFA signaling [105]; although they demonstrated that JMJD2D promotes the transcription of HIF1β by binding to its promoter to demethylate H3K9me3 and H3K36me3, the role of H3K36me3 remains to be determined as H3K36me2/3 is generally associated with active transcription. AML is one of the fatal cancers, and its overall survival is very poor, especially in patients with high expression of JMJD2D [106]. Myeloid cell leukemia-1 (MCL-1), an anti-apoptotic factor belonging to the Bcl-2 family, is essential for the progression of AML [111,112]. JMJD2D can elevate the survival of AML cells by transcriptionally activating the expression of MCL-1 [106]. JMJD2D is also identified as a prognostic marker for clear cell RCC (ccRCC), as the patients with low JMJD2D expression have a longer survival time after surgery, and pharmacological inhibition of JMJD2D can suppress the proliferation and angiogenesis of ccRCC both in vitro and in vivo by reducing the expression of JAG1 and VEGFR3 [107]. Intriguingly, Yao and colleagues reported that JMJD2D is a tumor suppressor in ESCC [108], indicating that the oncogenic role of JMJD2D is cancer-type dependent.

## 5. Targeted Cancer Therapy against JMJD2D

Currently, the strategies for competing Fe^2+^ or 2-OG are employed by the majority of JMJD2D small molecule inhibitors, due to the property of JMJD2D as a Fe^2+^/2-OG-dependent oxygenase [32]. To date, a variety of small molecule inhibitors have been reported to suppress the demethylation activity of JMJD2D, including JIB-04 [113], tripartin [114], 1.3-Amino-4-pyridine carboxylate derivatives [115], iridium (III) complex 1 [116], KDM4D-IN-1 [107,117], IOX1 (5-c-8HQ) [35,56,65,66,67,91,118], and 24s compound [119].

JIB-04 is a pan-selective JmjC demethylase inhibitor with IC50 of 290 nM for JMJD2D, but it is not a competitor of 2-OG [113]. JIB-04 can reduce cell viability and diminish the growth of H358 and A549 xenograft tumor in mice; however, the contribution of JMJD2D inhibition remains unclear [113]. Tripartin and Iridium (III) complex are reported to be the novel compounds to target JMJD2 enzymatic activity [114,116]. Although these compounds can inhibit the methylation of H3K9, their biological verification on tumor suppression is insufficient.

KDM4D-IN-1 is a highly specific inhibitor of JMJD2D with IC50 of 0.41 ± 0.03 μM. Yan et al. reported that the proliferation and angiogenesis of RCC cells both in vitro and in vivo are suppressed using the KDM4D-IN-1 inhibitor, which successfully inhibits the demethylation activity of JMJD2D [107]. IOX1 (also known as 5-c-8HQ) is a potent competitor of 2-OG and a broad-spectrum inhibitor of JmjC demethylases, including JMJD1A, JMJD2A, JMJD2C, JMJD2D, and JMJD3. Yu’s group reported that 5-c-8HQ can stall the progression of CRC and HCC by inhibiting the demethylase activity and the protein level of JMJD2D [35,65,66,91]. Peng et al. reported that 5-c-8HQ can suppress the proliferation of CT26, HCT116, and SW480 cells in vitro in a dose-dependent manner, and the protein levels of JMJD2D and Wnt/β-catenin target genes are also attenuated [35]. CRC-bearing mice treated with 5-c-8HQ have slower tumor growth; while given 5-c-8HQ, AOM/DSS-treated mice and *Apc*^min/+^ mice also develop fewer and smaller tumors in colons [35]. Furthermore, 5-c-8HQ can inhibit the expression of glycolysis-related genes by suppressing JMJD2D in CRC [65]. Deng et al. reported that 5-c-8HQ can inhibit the self-renewal abilities of Hepa1-6, Huh-7, and HepG2 cells in vitro in a dose-dependent manner; EpCAM and SOX9 proteins are also suppressed in these cells. The orthotopic graft tumor growth of Hepa1-6 is also inhibited by using 5-c-8HQ [91].

The strategy of combining multiple drugs is widely available for clinical tumor treatment. Vismodegib can suppress CRC progression by targeting smoothened, frizzled class receptor (SMO) of the Hedgehog pathway, and it has been approved for treating basal cell carcinoma. Zhuo et al. found that the proliferation of HCT116 cells co-treated with vismodegib and 5-c-8HQ is lower than that of monotherapy with either agent, and CRC-bearing mice given vismodegib and 5-c-8HQ together have more notable tumor inhibition, indicating that 5-c-8HQ can synergistically enhance the oncotherapy of vismodegib [66]. Of note, 5-c-8HQ can inhibit the expression of PD-L1 in CRC cells both in vitro and in vivo by suppressing JMJD2D and JAK-STAT3 signaling, and targeted inhibition of JMJD2D using 5-c-8HQ synergistically enhances the anti-tumor treatment of anti-PD-L1 immunotherapy [67]. Therefore, these studies indicate that JMJD2D is a novel molecular target for oncotherapy, especially in adjuvant tumor immunotherapy. However, no JMJD2D-specific blocking drugs are currently clinically available. Recently, Fang et al. identified a compound 24s, which has good selectivity targeting JMJD2D and significantly suppresses the proliferation and migration of CRC cells [119], and may have the potential to be developed to a specific JMJD2D inhibitor or a JMJD2D-specific targeting drug.

## 6. Conclusions

Previously, JMJD2D was recognized as an epigenetic multiplayer coordinating multiple biological activities, including DNA damage repair, DNA replication, cell cycle regulation, and inflammation. Recently, the oncogenic role of JMJD2D in multiple malignant tumors has been recognized and their potential carcinogenic mechanism has been preliminarily clarified (Figure 5). Briefly, JMJD2D promotes the transcription of various oncogenes by cooperating with β-catenin, Gli1/2, HIF1α, TCF4, and NICD, and then activates matched carcinogenic pathways, including Wnt/β-catenin, Hedgehog, mTOR-HIF1α, β-catenin/TCF4, and Notch signaling pathways that drive CRC or HCC progression. In addition, JMJD2D upregulates PD-L1 expression in CRC cells by activating JAK-STAT3 signaling to exert its antagonistic function against CD8^+^ T cell surveillance. JMJD2D can bind to p53 to block its transcriptional activity, thereby reducing the expression of p21 and PUMA tumor suppressors in HCC. JMJD2D contributes to hepatic fibrosis progression by promoting TLR4 transcription through H3K9me2/3 demethylation and then activates TLR4/NF-κB signaling in HSC. Furthermore, JMJD2D facilitates the progression of GIST by upregulating HIF1β expression. JMJD2D is crucial for the pathogenesis of AML and ccRCC, in which upregulation of MCL-1 and JAK1 is a major contributor. In inflammation, JMJD2D can be transcriptionally upregulated by the activated TNFα-NF-κB signaling, and both Trabid and TRIM14 can elevate the protein level of JMJD2D by reducing its ubiquitination; these may be the key to inflammation-induced tumors.

## Figures and Tables

**Figure 1 cancers-14-02841-f001:**
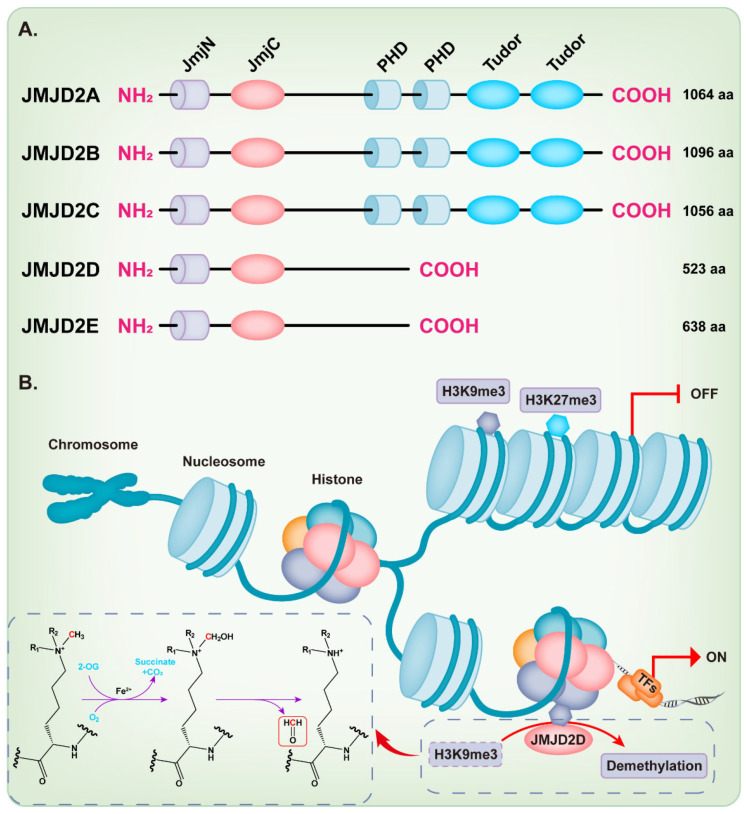
Structure and function of the JMJD2 histone demethylase subfamily. (**A**) The domain structure diagram of the JMJD2 histone demethylase subfamily. (**B**) JMJD2D facilitates a dioxygenase reaction requiring Fe^2+^, O^2^, and 2-oxoglutarate (2-OG) to demethylate H3K9me3.

**Figure 2 cancers-14-02841-f002:**
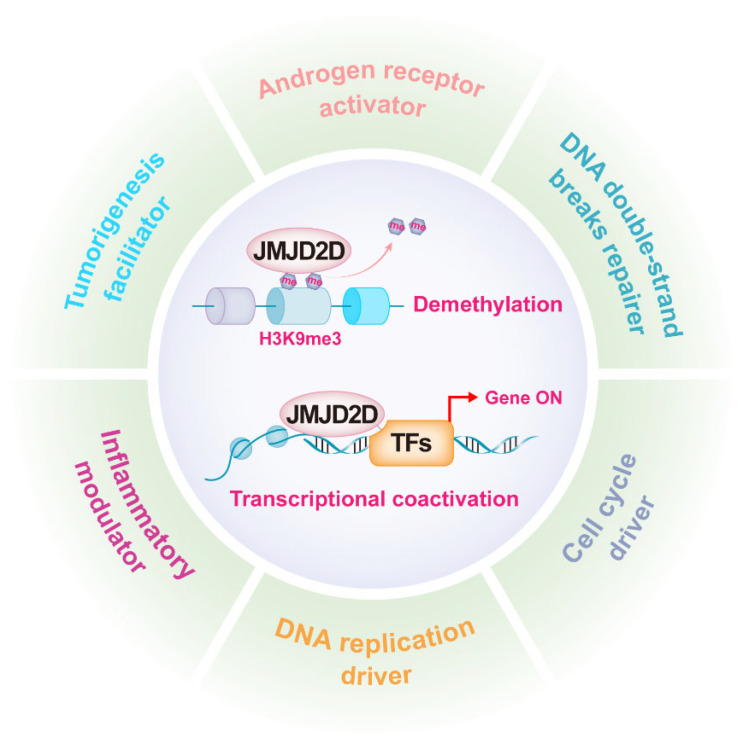
JMJD2D is a multifunctional epigenetic regulator. JMJD2D can promote gene transcription by antagonizing H3K9 methylation and cooperating with multiple transcription factors. JMJD2D has a variety of biological functions, including AR activation, DNA damage repair, DNA replication, cell cycle regulation, inflammation modulation, and tumorigenesis promotion.

**Figure 3 cancers-14-02841-f003:**
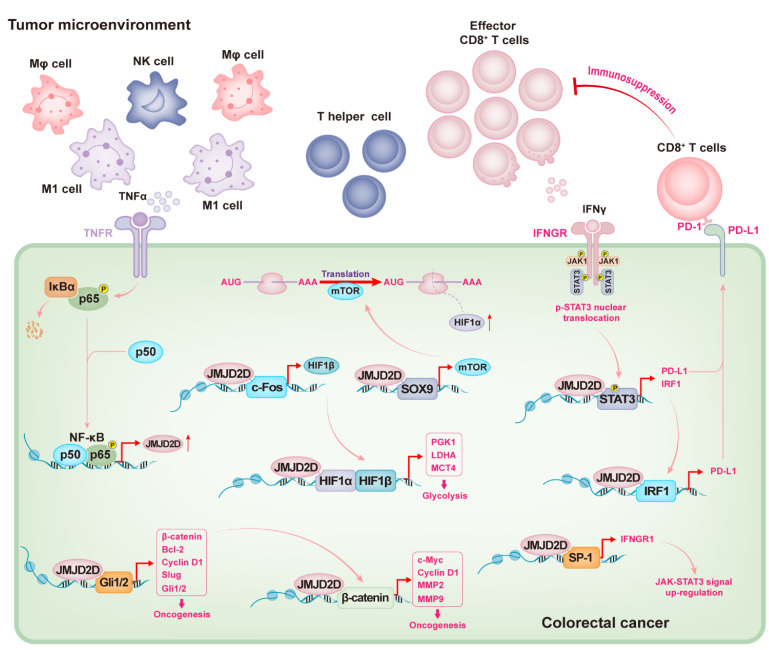
Overview of the mechanisms by which JMJD2D promotes CRC progression. Intestinal inflammation can trigger immune cells to release TNFα to upregulate JMJD2D in colorectal cells by activating NF-κB signaling. Highly expressed JMJD2D cooperates with Gli1/2 and β-catenin to promote the expression of various oncogenes, resulting in tumorigenesis. JMJD2D cooperates with c-Fos and SOX9to promote the expression of HIF1α and HIF1β, respectively. Subsequently, JMJD2D serves as a coactivator to promote the transcription of tumor glycolysis-related genes by elevating the HIF1α/β signaling. Again, JMJD2D cooperates with SP-1 to promote the expression of IFNGR1, which helps to maintain the signal that IFNγ triggers the upregulation of PD-L1 through the JAK-STAT3 pathway. Importantly, JMJD2D can cooperate with STAT3 and IRF1 to promote PD-L1 transcription. Subsequently, tumor cells antagonize the activation and effector functions of CD8^+^ T cells by activating PD-1/PD-L1 inhibitory signals through highly expressed PD-L1.

**Figure 4 cancers-14-02841-f004:**
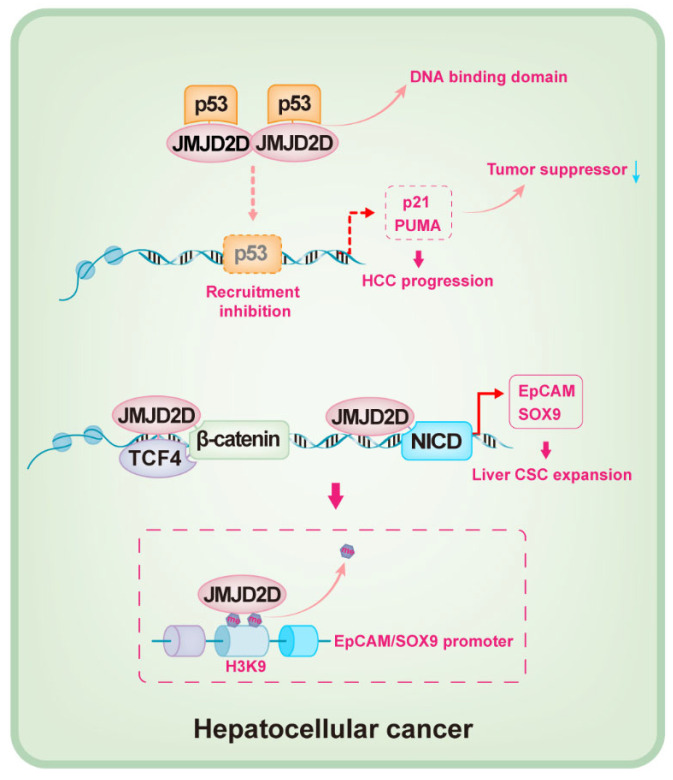
Schematic mechanisms by which JMJD2D promotes HCC progression. JMJD2D promotes HCC initiation and progression by directly binding to the DNA binding domain of p53 to inhibit the recruitment of p53 to the promoters of p21 and PUMA, resulting in the reduction of p21 and PUMA expression. JMJD2D promotes the self-renewal of LCSCs by cooperating with β-catenin/TCF4 and NICD to enhance the transcription of EpCAM and SOX9, respectively.

**Figure 5 cancers-14-02841-f005:**
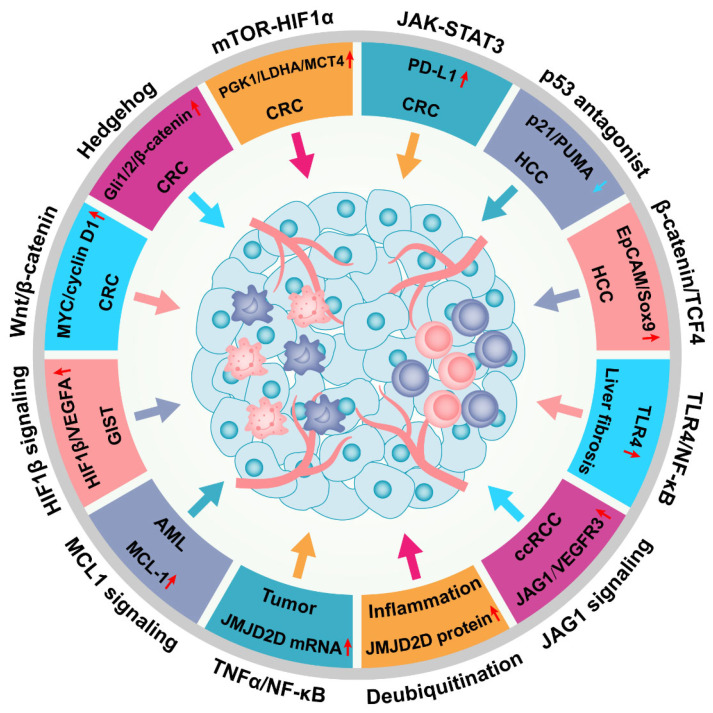
JMJD2D facilitates the progression of multiple malignant tumors by activating a variety of signaling pathways. JMJD2D promotes the progression of CRC by activating Wnt/β-catenin, Hedgehog, HIF1, and JAK-STAT3 signaling; and JMJD2D serves as a pivotal facilitator in the pathogenesis of HCC through antagonizing p53 or elevating the transcription of EpCAM and SOX9. Furthermore, JMJD2D is crucial for the pathogenesis of GIST, AML, ccRCC, and liver fibrosis. In inflammation, JMJD2D can be transcriptionally upregulated by the activated TNFα-NF-κB signaling, while both Trabid and TRIM14 can elevate the protein level of JMJD2D by reducing its ubiquitination.

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
