# Peer review of "Histone Demethylase JMJD2D: A Novel Player in Colorectal and Hepatocellular Cancers"

_cancers, 2022, doi:10.3390/cancers14122841_

Round 1
Reviewer 1 Report
Regulation of histone post-translational modifications (PTMs) is a critical mechanism involved in multiple cellular processes. Dysregulation of histone PTMs is associated with multiple pathologies, including different types of cancers. Chen et al reviewed the current known roles of the histone demethylase JMJD2D, a H3K9me3 demethylase, in colorectal and hepatocellular cancers. This is a well written review, which describes in details the different proteins interacting with JMJD2D and their effects in these two types cancers.
I have only a few minor comments:
- Lines 81-82 and Figure 1B: H3K79me2 and me3 are also associated with active transcription so the text and figure will need to be modified.
- The authors should add a figure summarizing the part 3 of the review (3. JMJD2D has multiple biological functions).
- Lines 345-347: While demethylation of H3K9me3 by JMJD2D to promote HIF1b expression is making sense, demethylation of H3K36me3 makes less sense to me as H3K36me3 is clearly associated with active transcription and usually found at low level on promoters and at the highest level in the gene body/3’end of the genes. Looking at ref 105, their chromatin immunoprecipitation (ChIP) experiments on H3K9me3 and H3K36me3 have not been controlled with total H3 level (and not performed with qPCR) so that will limit the claim of the authors of Ref 105.
- JMJD2D seems to interact with a wide range of proteins, especially transcriptional regulators. Is it known whether these interactions are mediated via the same domain or different domain of JMJD2D?
Reviewer 2 Report
This review is well organized and documented. I have only a suggestion:
The Authors should comment in a more exhaustive manner the Figure 4 of the conclusion. Among several molecules targeted by JMJD2D, some are direct others indirect targets of the demethylase.
In addition: pag. 2 line76 “Shim” should be written with capital S.
Author Response
Point 1: The Authors should comment in a more exhaustive manner the Figure 4 of the conclusion. Among several molecules targeted by JMJD2D, some are direct others indirect targets of the demethylase.
Re: Thank you for your careful review and constructive assistance. We have supplemented the exhaustive text to summarize the conclusions of Figure 4 according to your suggestion, and the modification has been marked up using the “Track Changes” function.
Point 2: In addition: pag. 2 line76 “Shim” should be written with capital S.
Re: Thank you for your careful review, we have revised it.